# Characterization of the *1-Deoxy-D-xylulose 5-Phosphate synthase* Genes in *Toona ciliata* Suggests Their Role in Insect Defense

**DOI:** 10.3390/ijms24032339

**Published:** 2023-01-25

**Authors:** Yueyang Wang, Yue Li, Huiyun Song, Zhi Wang, Pei Li

**Affiliations:** 1College of Forestry and Landscape Architecture, South China Agricultural University, Guangzhou 510642, China; 2Guangdong Key Laboratory for Innovative Development and Utilization of Forest Plant Germplasm, Guangzhou 510642, China

**Keywords:** *Toona ciliata* Roem., *TcDXS*, *Hypsipyla robusta* stress, bioinformatics analysis, expression pattern

## Abstract

The first enzyme, *1-Deoxy-D-xylulose-5-phosphate synthase* (*DXS*), in the 2-C-methyl-D-erythritol-4-phosphate (MEP) pathway for isoprenoid precursor biosynthesis has been reported to function differently according to species. However, the current state of knowledge about this gene family in *Toona ciliata* is limited. The *TcDXS* gene family was identified from the whole genome of *T. ciliata* by firstly using bioinformatics analysis. Then, the phylogenetic tree was built and the promoter cis-elements were predicted. Six *DXS* genes were identified and divided into three groups, which had similar domains and gene structure. They are located on five different chromosomes and encode products that do not vary much in size. An analysis of the cis-acting elements revealed that *TcDXS* genes possessed light, abiotic stress, and hormone responsive elements. Ultimately, *TcDXS1/2/5* was cloned for an in-depth analysis of their subcellular localization and expression patterns. The subcellular localization results of *TcDXS1/2/5* showed that they were located in the chloroplast envelope membranes. Based on tissue-specific analyses, *TcDXS1/2/5* had the highest expression in mature leaves. Under *Hypsipyla robusta* stress, their different expressions indicated that these genes may have insect-resistance functions. This research provides a theoretical basis for further functional verification of *TcDXSs* in the future, and a new concept for breeding pest-resistant *T. ciliata*.

## 1. Introduction

*Toona ciliata* belongs to the *Toona* subgenus of the Meliaceae family. It grows fast, with straight trunks, and produces a special color wood with a beautiful grain [1]. It is an endangered plant with key protection status in China, where it has high economic value and development prospects as well as the reputation of being “Chinese *mahagoni*” [2]. *T. ciliata* is distributed in southern China and tropical and subtropical regions [3,4]. However, the harm that *Hypsipyla robusta* Moore causes to *T. ciliata* cannot be underestimated. *H. robusta* eat the young shoots of *T. ciliata*, causing it difficulties in growing normally or even leading to death, and making it difficult for the trees to grow wood, which seriously affects the technological value of the wood [5]. Therefore, it is urgent to control *H. robusta*. Previous studies have shown that *H. robusta* can be managed by physical, chemical, and biological methods, but these are time and energy consuming [6]. With the gradual maturation of molecular breeding technology, another way to decrease the harm caused by pests is the cultivation of insect-resistant *T. ciliata* using this technology.

Terpenoids are the most diverse secondary metabolites with many volatile components [7]. Terpenoids are natural compounds, categorized according to the number of C_5_ isoprene units in the molecular formula, which can be divided into hemiterpenes (C_5_), monoterpenes (C_10_), sesquiterpenes (C_15_), diterpenes (C_20_), sesterterpenes (C_25_), triterpenes (C_30_), and tetraterpenes (C_40_)) [8]. There are two pathways for the synthesis of terpenes, namely, the 2-C-methyl-derythritol4-phosphate (MEP) pathway [9] and the mevalonate (MVA) pathway [10]. In these two pathways, the biosynthesis of terpenoids includes three stages: firstly, the precursors dimethyl allyl pyrophosphate (DMAPP) and isopentenyl diphosphate (IPP) are formed, followed by a synthesis of the initial terpenoid products; then, the last stage is the formation of the final product under the action of various modifying enzymes [11]. Typically, IPP and DMAPP, which are necessary for the synthesis of monoterpenes, diterpenes and tetraterpenes, are produced by the MEP pathway, while the synthesis of sesquiterpenes, sterols, triterpenes and their diosgenin derivatives are produced by the MVA pathway. [12]. Terpenoids are related not only to plant growth, development and resistance to the external environment but also to the volatile monoterpenes synthesized and released in flower organs and leaves, which play a signaling role between plants and the environment [13]. For example, linalool and (*E*)-*β*-ocimene have the strongest attraction to the female *Aphidius ervi* [14]. Monoterpenes are volatile and often exist in the form of volatile oil, which can attract pests. Therefore, from this perspective, we hope to provide a way to solve the damage caused by *H. robusta*.

The preliminary results of our research group showed that a number of diferentially expressed genes in the terpenoid biosynthesis pathway were screened through transcriptome analysis of 3-month-old (non-harmful stage) and 2-year-old (severely harmful stage) leaves of *T. ciliata*. The most significant difference in expression level is the important rate-limiting enzyme *1-deoxy-D-xylose-5-phosphate synthase* (*DXS*) gene [15]. It is generally believed that the *DXS* gene plays a key regulatory role in catalyzing the synthesis of terpenoid precursors, affecting the proportion of final products produced by other terpenoids [16,17]. Therefore, it is necessary to understand the *TcDXS* gene family in-depth.

According to reports, the *DXS* gene family members of numerous plants were cloned and discovered, and most plants carry more than one *DXS* gene. For instance, the majority of plants in 16 species of Solanaceae contain three *DXS* genes, as do *Arabidopsis thaliana*, *Oryza sativa*, *Zea mays*, *Artemisia annua*, and *Morus notabilis*, although *Solanum habrochaites* and *Withania somnifera* only carry one *DXS* gene and *Nicotiana tabacum* has the most *DXS* genes. They also have several similar domains and motifs, which means the *DXS* gene family is highly conserved [17,18,19,20,21]. It is also essential to the terpene biosynthesis of plants [22]. For example, *OsDXS2* can provide precursors to carotenoid accumulation in *O. sativa* seeds [18], while the *MnDXS2A/2B* genes can regulate the synthesis of terpenoids, which has something to do with plant defense against herbivores in *M. notabilis* [17]. The *AaDXS2* gene may be related to the artemisinin biosynthesis of *A. annua* [20]. Overexpression of the *DXS* gene can enhance the accumulation of terpenoid secondary metabolites [23]. In *A. thaliana*, the heterologously overexpressed *PmDXS2* gene of *Pinus massoniana* increases the content of chlorophyll and carotenoids [24]. Furthermore, *DXS* gene expression varies in different parts of plants, which is impacted by the circadian rhythm and other factors [25,26]. According to existing phylogenetic analysis, *DXS* genes are divided into three groups: group I genes are housekeeping genes [27]; group II genes can encode plant-specific secondary metabolites, which may be related to the defense response and isoprene synthesis [21]; and group III genes are slightly longer than those of group I and group II, which may be related to postembryonic development and reproduction [28].

In this report, we identified the *DXS* gene family of *T. ciliata* and cloned *TcDXS* genes from group II. The gene structure, conserved domain, promoter prediction, expression pattern, and subcellular localization, were further analyzed. Thus, this study presents an important basis upon which to investigate the function of *TcDXS* genes in *T. ciliate*, and the cultivation of excellent insect-resistant varieties of *T. ciliata*. It also has theoretical significance and potential application value for the sustainable development of *T. ciliata*.

## 2. Results

### 2.1. Identification and Characterization

To identify the *TcDXS* gene family of *T. ciliata*, the protein sequences of AtDXS of *A. thaliana* were used by BLASTP. The identified conserved domain of TcDXS proteins were verified by the PFAM database, and six TcDXS proteins of *T. ciliata* were identified. The six *TcDXS* genes were renamed, changing from *TcDXS1* to *TcDXS6*, according to their distribution on different chromosomes (Table 1).

In addition, the coding sequence (CDS) length, isoelectric point (pI), molecular weight (MW), and amino acid composition of six *TcDXSs* proteins were analyzed by ExPASy-ProtParamtool (https://web.expasy.org/protparam/, accessed on 16 February 2022). Except for *TcDXS2* and *TcDXS6*, the pIs of the other *TcDXSs* were greater than seven; these were considered to be basic proteins. Subcellular localization predictions revealed that all *TcDXSs* were in the chloroplast.

### 2.2. Multiple Sequence Alignment and Phylogenetic Analysis

The results of multiple sequence alignments of AtDXSs and TcDXSs protein sequences showed that they are highly conserved (Appendix A). *TcDXS* amino acid sequences include the transket-pyr binding site GDGA(X)TAG-QAYEA(X)NNAGFLD(X)N(X)IV(X)LNDN and the transketolase-C binding site AGLVG(X)DGPTHCGAFDITYM(X)CLPNMVVMAPSD [29]. In addition, the *TcDXS1* to *TcDXS5* amino acid sequences of *T. ciliata* and *AtDXS1* and *AtDXS2* amino acid sequences contained the highly conserved special domain “DRAG” of the DXS family, but the *TcDXS6* amino acid sequence and *AtDXS3* amino acid sequence contained a special domain, “TSAG” [30].

The amino acid sequences of *TcDXS* proteins of *T. ciliata* and other plants were used to construct an unrooted phylogenetic tree using MEGA7.0 software using the neighbor-joining method (Figure 1). The DXS protein sequences included *A. thaliana*, *G. biloba*, *H. brasiliensis*, *M. truncatula*, *O. sativa*, *P. trichocarpa*, *R. communis*, *T. sinensis*, and *Z. mays*.

According to the results of the unrooted phylogenetic tree, the DXS gene family was divided into groups I, II and III; *TcDXS3* and *TcDXS4* have high homology with *A. thaliana AtDXS1* and *AtDXS2* as well as *DXS1* of other species, which belong to group I and are housekeeping genes [27]. *TcDXS1*, *TcDXS2* and *TcDXS5* share high homology with *OsDXS2*, *PtDXS2* and *PtDXS3*, and *DXS2* of other species, which belong to branch group II of the *DXS* genome [31]. This can encode plant-specific secondary metabolites, which may be related to the defense response and isoprene synthesis [21]. *TcDXS6* has high homology with *PtDXS4*, *OsDXS3*, *AtDXS3*, *RcDXS3* and *TsDXS5*; it belongs to group III, which may be related to postembryonic development and reproduction [28].

### 2.3. Chromosome Location, Gene Structure, and Conserved Domain Analysis

The *TcDXSs* were located on Chr11, Chr12, Chr21, Chr22 and Chr25, and there were two *TcDXSs* genes on Chr25 (*TcDXS5* and *TcDXS6*) (Figure 2). Gene structure analysis indicated that the same family members have similar distribution patterns. Among them, *TcDXS6*, classified as subfamily III, has a longer intron than the other genes (Figure 3A). When using PFAM and TBtools to predict the conserved domain of the *TcDXSs* (Figure 3B), the results showed that all of the *TcDXSs* contained three domains, DXP-synthase-N, Transket-pyr, and Transketolase-C. Then, using MEME to predict the motifs of the *TcDXSs*, each *TcDXS* contained from 9 to 10 motifs (Figure 4A). The majority motifs were shared by all members; only one gene was different from the others. For example, only subfamily III member *TcDXS6* does not have motif eight. The motifs’ sequences are shown in Figure 4B.

### 2.4. TcDXSs Promoter cis-Acting Element Prediction

The promoter cis-acting elements of *TcDXSs* genes were related to abscisic acid responsiveness, light responsiveness, defense and stress responsiveness, and other functions (Figure 5). The elements related to light responsiveness were abundant, including ATC-motif, Box4, TCCC-motif, etc., among which Box4 was the most abundant (Figure 6).

### 2.5. Expression Patterns under H. robusta Stress

The CDSs of the *TcDXS1*, *TcDXS2* and *TcDXS5* genes were used to determine part of the intron and exon structure of the gene by NCBI-Blast; then, reverse transcription-quantitative PCR (RT-qPCR) primers were designed across the introns. Finally, primers with good specificity, no primer dimer, and unimodal melting curves were selected for the RT-qPCR (Appendix A).

Reverse transcription-quantitative PCR was used to detect the expression of three *TcDXS* genes in young stems and leaves under *H. robusta* stress (Figure 7). The results showed different expression trends in young stems and leaves with increasing stress duration. In young stem tissue, the relative expression of *TcDXS1* and *TcDXS5* first increased, then decreased, and then increased. The relative expression of *TcDXS1* was the highest after 21 h of stress, while that of *TcDXS5* was the highest after 3 h of stress. The relative expression levels of *TcDXS2*, *TcDXS1* and *TcDXS5* were different and showed a gradual upward trend, and the relative expression was the highest after 21 h of stress. In leaf tissue, the relative expression levels of the three genes showed different trends. Among them, the relative expression of *TcDXS1* fluctuated and was the lowest, and was the highest after 12 h of stress. The expression of *TcDXS2* decreased but was higher than the original expression after 21 h of stress. The expression of *TcDXS5* gradually decreased after 21 h of stress, and the expression was the lowest after 21 h of stress, while the relative expression of *TcDXS5* first increased and then decreased, peaked at 12 h, and then rapidly decreased.

### 2.6. Expression Patterns of Different Tissues and Provenances

The relative expression of *TcDXS1/2/5* were determined in different tissues and provenances of *T. ciliata* (Figure 8). The results showed that the expression of *TcDXS1/2/5* were all expressed at the highest level in mature leaves. In different provenances of *T. ciliata*, the expression of *TcDXS1* and *TcDXS2* were the highest in P7 (Tianlin Guangxi), while *TcDXS5* was the highest in P4 (Lechang Guangdong).

### 2.7. Subcellular Location of TcDXS Proteins

Three TcDXS genes (TcDXS1, TcDXS2 and TcDXS5) related to monoterpene biosynthesis were selected for cloning. The results of 1.5% agarose nucleic acid gel detection are shown in Appendix A; specific bands of the expected size are amplified. The ORFs of the three genes are consistent with their CDSs. To validate the predicted subcellular localization of TcDXS proteins, the constructed TcDXSs fragments, without stop codons, were inserted into the pCAMBIA1300 vector to fuse these genes with the 35S promoter-driven GFP protein to produce C-terminal GFP fusions, when expressed in *A. thaliana* protoplasts. The subcellular localization results showed that the GFP fused to the TcDXS proteins localized in the chloroplast envelope membrane (Figure 9).

## 3. Discussion

*H. robusta* is a low-grade dangerous pest in China [32], but the harm caused to *T. ciliata* should not be underestimated. The prevention and control of insect pests includes not only physical and chemical controls but also a series of effective defense mechanisms. Plants usually enhance their ability to resist pests by producing nutrients, secondary metabolites, defense enzymes, and plant hormones [33].

Secondary metabolites are the result of interactions between plants and biotic and abiotic factors in the process of long-term evolution [34]. Terpenes, as secondary metabolites, play a very important role in the antagonistic and mutually beneficial relationships between organisms. They can protect many types of plants, animals and microorganisms from predators, pathogens and competitors. They also participate in conveying messages about food, mates, and enemies among organisms [35]; for example, monoterpenes in pine resin can reduce the damage of *Dendroctonus valens* to *Pinus tabulaeformis* [36].

Previous research has demonstrated that the presence and concentration of the monoterpene α-pinene, in particular its ratios with other substances, is very likely to operate as a signal material for *T. ciliata* to remove *H. robusta* [37,38]. Further, the *DXS* genes were screened through *T. ciliata* transcriptome analysis [15]. These are essential for the synthesis of monoterpenoid precursors in the MEP pathway. Therefore, the *DXS* genes were studied in this study.

*O. sativa* and *A. annua* both contain three *DXS* genes, which were split into three branches by the evolutionary tree. *A. thaliana* also has three *DXS* genes; however, they were divided into two groups (I and III) [18,20]. In *Santalum album*, *SaDXS1A* and *SaDXS1B* were found and divided in the first branch of the phylogenetic tree [39]. In *M. notabilis*, *MnDXS1* was found in the first branch and *MnDXS2A* and *MnDXS2B* were found in the second branch of the phylogenetic tree [17]. Additionally, research on *DXS* genes in several Solanaceae plants revealed that the genes are conserved; they can be classified into three groups and the role of *DXS* was different in each branch [21]. In this study, according to the whole genome sequence information of *T. ciliata*, six *TcDXS* genes were identified and screened, and the number of *DXS* genes may be different from that in other plants including dicotyledons (*A. thaliana*, *Croton stellatopilosus*, *P. massoniana* and *A. annua*) [20,24,29,40] and monocotyledons (*O. sativa* and *Z. mays*) [18,19]. The *TcDXS* genes can be divided into three subfamilies, *TcDXS3* and *TcDXS4* belong to the *DXS* subfamily group I and are housekeeping genes [41]. *TcDXS1*, *TcDXS2* and *TcDXS5* belong to group II of the DXS subfamily and are related to the plant’s secondary metabolism [42]. *TcDXS6* belongs to group III of the DXS subfamily, but the function of the enzyme encoded by *T. ciliata* is not clear and may be related to the genes of postembryonic development and reproduction [28]. The phylogenetic tree results indicated that these plant DXS genes were derived from one ancestor gene and developed into three branches, with the corresponding functions after the species diverged, which was consistent with previous reports on the *DXS* gene family.

Based on the analysis of the results of *Schizonepeta tenuifolia* and *Cinnamomum camphora*, the DXS proteins contained three domains, DXP-synthase-N, Transket-pyr, and Transketolase-C [30,43]. The TcDXS proteins of *T. ciliata* have three domains and two binding sites, which is consistent with previous studies. According to the function determined by the protein domains, TcDXS proteins were inferred to have the same function as other plant DXS proteins in regulating the synthesis of secondary metabolites.

The promoter is an important element of gene expression regulation, and the correct regulation of gene expression by the promoter requires the synergy of the core promoter and the upstream and downstream cis-acting elements [44]. G-box, ACE, Box 4, TCCC-motif, GATA-motif, ATCT-motif, GT1-motif and other light-responsive cis-acting elements have been reported [45,46,47]. In soybean, the promoter of *GmPLP1* contains ABRE, ERE, LTR, G-box, CGTCA-motif, GT1-motif, ATCT-motif, Box4, Sp1, I-box, and the TCT-motif, which can enhance gene expression under plant growth regulators and light stress [48]. The promoter cis-acting element prediction for the six genes was mainly focused on ABRE, Box4 and G-Box of the light-responsiveness-related cis-acting element, which can be further used to investigate plant stress-response patterns.

In terms of their expression and regulation, the expression of the *DXS* gene is related to various factors, such as the different parts of plants and their circadian rhythm; there are also tissue differences [25,26]. In *Andrographis paniculata*, the expression of the *DXS* gene was strongest in the leaves, followed by the stems and roots [49]. In *Pelargonium hortorum*, *DXS* had the highest expression in early developmental leaves [23]. In *M. notabilis*, *MnDXS2B* was mainly expressed in the roots, though *MnDXS1* and *MnDXS2A* had no tissue specificity [17]. The *DXS* gene was highly expressed in leaves in maize [19], and the highest expression of the *PtDXS* gene was found in *Populus trichocarpa* leaves [22].

The results of this study showed that the *TcDXS1*, *TcDXS2* and *TcDXS5* genes were mainly expressed in mature leaves of *T. ciliata*, followed by the roots and flowers, which was most consistent with the expression of *DXS* genes in *Populus trichocarpa*. In addition, there were differences among the three genes in terms of provenance. The expression patterns of the *TcDXS1*, *TcDXS2* and *TcDXS5* genes in different tissues of *T. ciliata* under pest stress were also analyzed. The results showed that the *TcDXS1*, *TcDXS2* and *TcDXS5* genes were upregulated under pest stress in the young stem tissue of *T. ciliata*. In leaves, the *TcDXS1* gene was only downregulated at 3 h under pest stress, while the *TcDXS2* and *TcDXS5* genes were inhibited under pest stress. It is speculated that the *TcDXS1*, *TcDXS2* and *TcDXS5* genes of *T. ciliata* positively regulate the stress response of *T. ciliata* insect pests in the young stems and negatively regulate the stress response of *T. ciliata* insect pests in the leaves.

## 4. Materials and Methods

### 4.1. Plant Materials and Treatments

Experimental materials were obtained from the Qilin North Experimental Base (SCAU, Guangzhou, China) of South China Agricultural University. Buds, young and mature leaves and young stems were gathered from 3-year-old *T. ciliata*. Leaves of different provenances were gathered from 5-year-old *T. ciliata* (Appendix A). *H. robusta* was caught and put on the 5-year-old *T. ciliata* seedlings; their young stems and leaves were collected after 0, 3, 12 and 21 h. Three biological replicates for each treated sample and all samples were snap-frozen in liquid nitrogen and stored at −80 °C.

### 4.2. TcDXS Gene Family of T. ciliata Identification and Analysis

Complete genome sequences and annotation information of *T. ciliata* (record number: CNP0001985) were downloaded from the CNGB (https://www.cngb.org/, accessed on 11 February 2022). The protein sequences of AtDXS in *A. thaliana* were obtained from TAIR (https://www.arabidopsis.org, accessed on 12 February 2022). Two rounds of BLASTP identified *TcDXSs*: the AtDXS1/2/3 protein sequences were used as the query to search for potential *TcDXSs* with an *E*-value of 1 × 10^−5^ by TBtools (Blast Compare Two Seqs [Sets] <Big File>) [50]. After this, potential *TcDXSs* protein sequences were submitted to PFAM with the Pfam-A database (*E*-value = 1 × 10^−5^) (http://pfam.xfam.org/, accessed on 12 February 2022) and Swiss-Port database in NCBI-BLASTP (https://blast.ncbi.nlm.nih.gov/Blast.cgi, accessed on 12 February 2022) to check if they contained all DXP-synthase-N (PF13292.6), Transket-pyr (PF02779.24) and Transketolase-C (PF02780.20) conserved domains. Next, the protein sequences of all *TcDXSs* were submitted to ExPASy-ProtParam (https://web.expasy.org/protparam/, accessed on 16 February 2022) and Plant-mPLo (http://www.csbio.sjtu.edu.cn/bioinf/plant-multi/, accessed on 16 February 2022) to predict the proteins’ physical and chemical properties and subcellular localization [51].

### 4.3. Multiple Sequence Alignment and Phylogenetic Analysis

Muscle with defaults in Jalview version 2.0 was used to calculate the multiple sequence alignments of *TcDXSs* [52] and edited with GeneDoc [53]. DXS protein sequences (Appendix A), including *A. thaliana*, *Ginkgo biloba*, *Hevea brasiliensis*, *Medicago truncatula*, *O. sativa*, *Populus trichocarpa*, *Ricinus communis*, *Toona sinensis*, and *Zea mays* homologous proteins, were selected and acquired from TAIR (*A. thaliana*) and NCBI. Then, ClustalW in MEGA7.0 software was used to perform the multiple sequence alignments, removing the gaps at both ends and replacing the gaps in the middle with ‘?’. MEGA7.0 was used to construct neighbor-joining (NJ) phylogenetic trees with 1000 bootstrap replicates [54]. The protein sequences of *AtDXSs*, *GbDXSs*, *HbDXSs*, *MtDXSs*, *OsDXSs*, *PtDXSs*, *RcDXSs*, *TsDXSs*, *ZmDXSs* and *TcDXSs* were used.

### 4.4. Chromosome Location, Gene Structure, and Conserved Domain Analysis

Analysis of the conserved domain of *TcDXSs* was realized through PFAM and the MEME program (https://meme-suite.org/meme/tools/meme, accessed on 12 February 2022). TBtools (Gene Location Visualize from GTF/GFF and Gene Structure View (Advanced)) was used to visualize the chromosome location, gene structure, and conserved domain [50].

### 4.5. Promoter cis-Acting Element Prediction

PlantCare was used to predict the promoter cis-acting elements [55]. The promoter was extracted with TBtools (Gtf/Gff3 Sequences Extract) from 2000 bp sequence upstream of the coding region of the *TcDXSs* genes and the results were visualized [50].

### 4.6. Gene Cloning

Primer 5 was used to design the *TcDXS1/2/5* primers (Appendix A). The template was the cDNA of *T. ciliata*, the PCR programs were predenaturation at 95 °C for 3 min; 35 cycles of denaturation at 95 °C for 15 s, annealing at 55 °C for 15 s and extension at 72 °C for 70 s; and extension at 72 °C for 5 min (Appendix A).

### 4.7. RNA Extraction and Reverse Transcription-Quantitative PCR (RT-qPCR) Analysis

With a HiPure HP Plant RNA Kit (Magen), the total RNA of all samples was extracted and the qualified high-quality samples were stored at −80 °C. cDNA was synthesized from 1 μg of total RNA according to the instructions of the HiScript II Reverse Transcription Kit (Vazyme).

The RT–qPCR primers were designed with NCBI across introns and detected by PCR (Appendix A). Using cDNA from the different tissues and provenances of *T. ciliata*, and young leaves and young stems under *H. robusta* stress for different times, as templates, the 20 µL reaction system included 2×ChemQ Universal SYBR qPCR Master Mix 10 µL, cDNA 2 µL, each primer (10 µM) 0.4 µL, and ddH_2_O 7.2 µL. The reaction procedure was as follows: 95 °C 30 s, 40 cycles of 95 °C 15 s, 60 °C 20 s and 72 °C 10 s. The melting curve was analyzed at 65–95 °C, and the specificity of the product was judged according to the peak diagram of the dissolution curve. There were three biological repeats in each sample and three technical repeats in each biological repeat, it was found that *TUB-α*, *HIS1*, *PP2C59* and *MUB* could be used as reference genes in those conditions [6]. Based on the obtained Ct values, the gene expression levels were calculated using the 2^−ΔΔCt^ method [56] in GraphPad Prism 8 software.

### 4.8. Vector Construction and Subcellular Localization Analysis

The pCAMBIA1300 vector and TcDXS1/2/5::GFP fusion vector containing the fluorescent signal of green fluorescent protein (GFP) were constructed and then transferred into the *Agrobacterium tumefaciens* GV3101 receptor state; the primers that were used are shown in Appendix A. Next, the empty pCAMBIA1300-GFP and expression vector pCAMBIA1300-TcDXSs-GFP GV3101 strains were infiltrated with a buffer on *A. thaliana* protoplasts [57], and the green fluorescence was detected by laser confocal microscopy.

## 5. Conclusions

Members of the *TcDXS* gene family of *T. ciliata* were identified, and the *TcDXS1*, *TcDXS2* and *TcDXS5* genes of *T. ciliata* were successfully cloned, with the expression patterns and subcellular localization of the three genes preliminarily characterized. This provided a theoretical basis for the identification of *TcDXS* function, and offers an opportunity for the cultivation of excellent insect-resistant varieties of *T. ciliata* in the future. It also provides a foundation upon which to further analyze the regulatory relationship of this important candidate gene, *TcDXS*, in the biosynthesis of the terpenoid MEP pathway in *T. ciliata*.

## Figures and Tables

**Figure 1 ijms-24-02339-f001:**
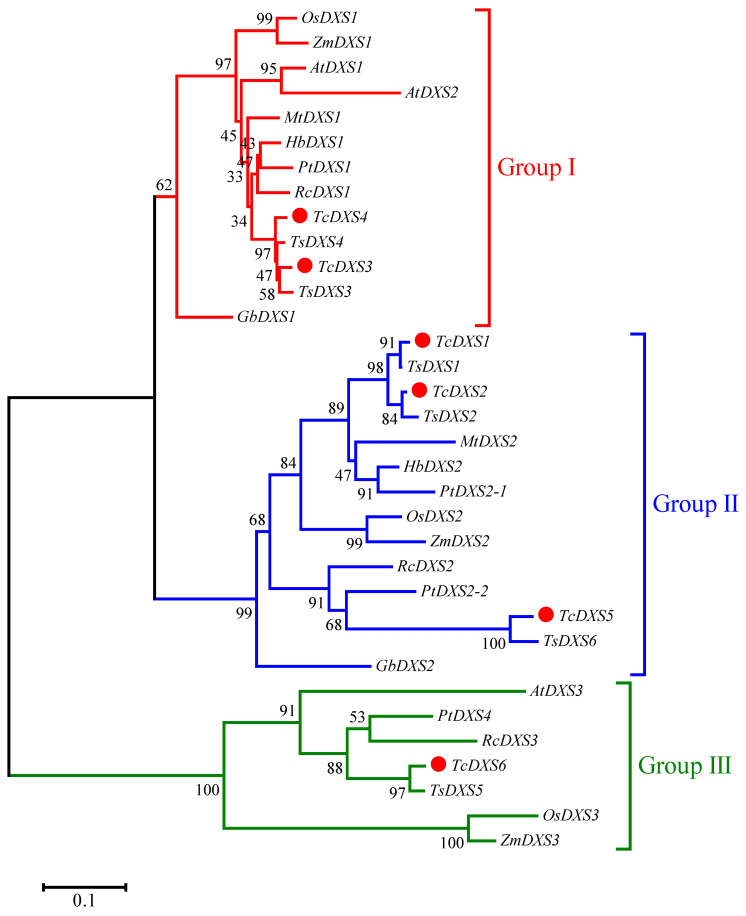
Phylogenetic tree of the DXS gene family from *T. ciliata*, *A. thaliana*, *G. biloba*, *H. brasiliensis*, *M. truncatula*, *O. sativa*, *P. trichocarpa*, *R. communis*, *T. sinensis*, and *Z. mays*. A neighbor-joining tree was constructed from DXSs’ protein sequences (Appendix A).

**Figure 2 ijms-24-02339-f002:**
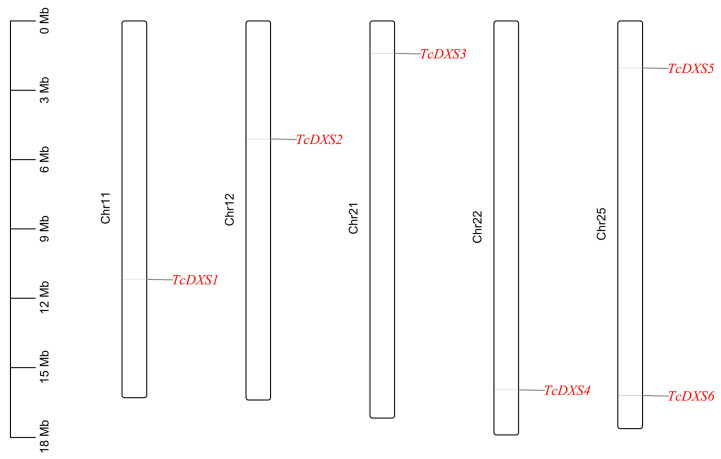
Chromosome location of the *TcDXSs* gene family in *T. ciliata*.

**Figure 3 ijms-24-02339-f003:**
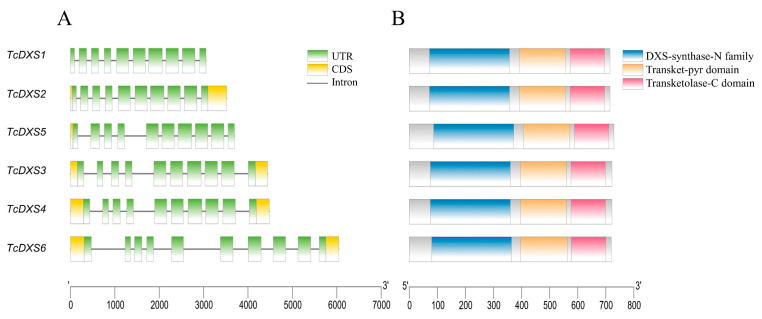
Gene structure and conserved domain of TcDXSs. (**A**) Analysis of the structure of the TcDXS genes. The green and yellow boxes represent UTRs and CDSs, respectively; black lines represent introns. (**B**) Analysis of the conserved domain of the TcDXS proteins. Green boxes indicate the DXS-synthase-N family, yellow boxes indicate the Transket-pyr domain and pink boxes indicate the Transketolase-C domain.

**Figure 4 ijms-24-02339-f004:**
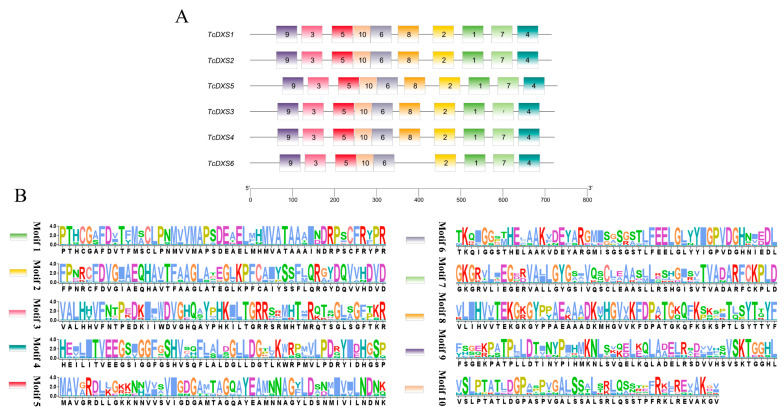
Motifs of TcDXS proteins. (**A**) Analysis of the motifs of TcDXS proteins by MEME. (**B**) Different colors represent different motifs, and the motif sequence is shown in the figure.

**Figure 5 ijms-24-02339-f005:**
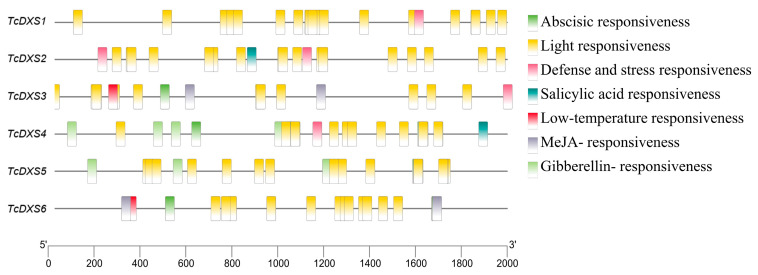
Analysis of the *TcDXS* promoters’ cis-acting elements, with seven response types; each containing several elements.

**Figure 6 ijms-24-02339-f006:**
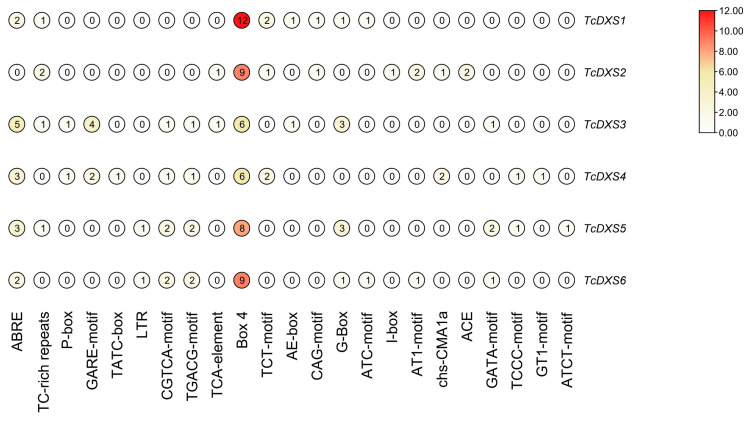
Numbers of each type of motif related to light responsiveness (Appendix A).

**Figure 7 ijms-24-02339-f007:**
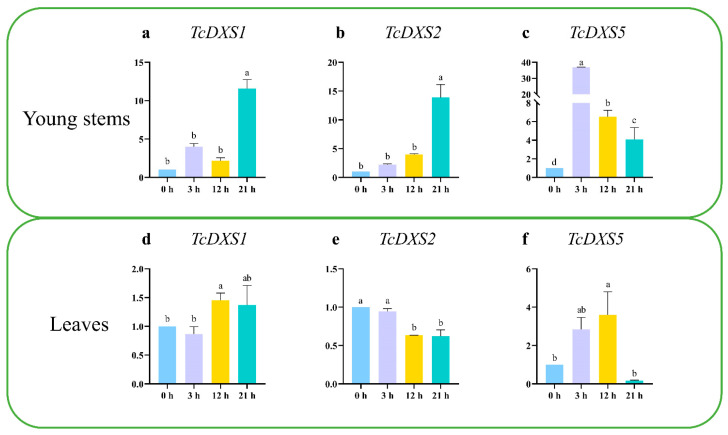
Expression patterns of *TcDXSs* genes during 0, 3, 12, and 21 h under *H. robusta* stress. The data represent the mean of three independent biological replicates relative to 0 h, different letters above the bars represent statistically significant differences (*p* < 0.05) detected by a one-way ANOVA test. (**a**) *TcDXS1* gene during 0, 3, 12, and 21 h under *H. robusta* stress in young stems. (**b**) *TcDXS2* gene during 0, 3, 12, and 21 h under *H. robusta* stress in young stems. (**c**) *TcDXS5* gene during 0, 3, 12, and 21 h under *H. robusta* stress in young stems. (**d**) *TcDXS1* gene during 0, 3, 12, and 21 h under *H. robusta* stress in leaves. (**e**) *TcDXS2* gene during 0, 3, 12, and 21 h under *H. robusta* stress in leaves. (**f**) *TcDXS5* gene during 0, 3, 12, and 21 h under *H. robusta* stress in leaves.

**Figure 8 ijms-24-02339-f008:**
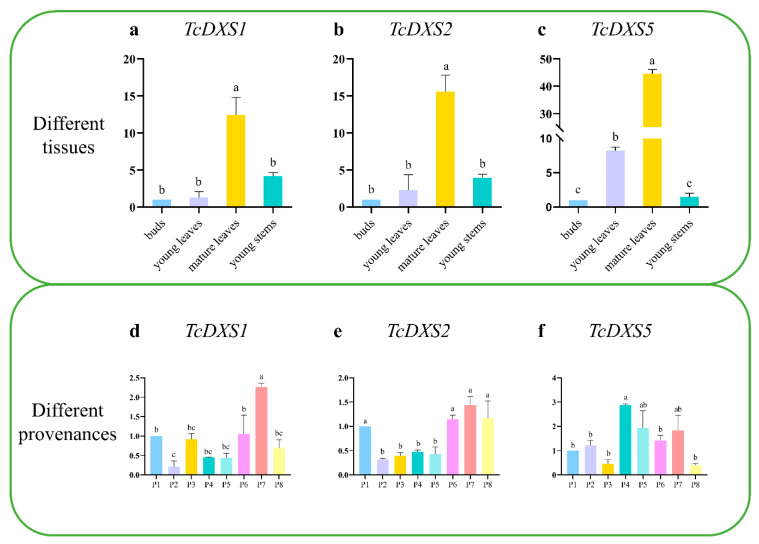
Expression of *TcDXS1*, *TcDXS2* and *TcDXS5* genes in different tissues and different provenances. The data represent the mean of three independent biological replicates relative to buds and P1 (Pupiao Yunnan), respectively. Different letters above the bars represent statistically significant differences (*p* < 0.05) detected by a one-way ANOVA test. (**a**) Expression of *TcDXS1* gene in different tissues. (**b**) Expression of *TcDXS2* gene in different tissues. (**c**) Expression of *TcDXS5* gene in different tissues. (**d**) Expression of *TcDXS5* gene in different provenances. (**e**) Expression of *TcDXS2* gene in different provenances. (**f**) Expression of *TcDXS5* gene in different provenances.

**Figure 9 ijms-24-02339-f009:**
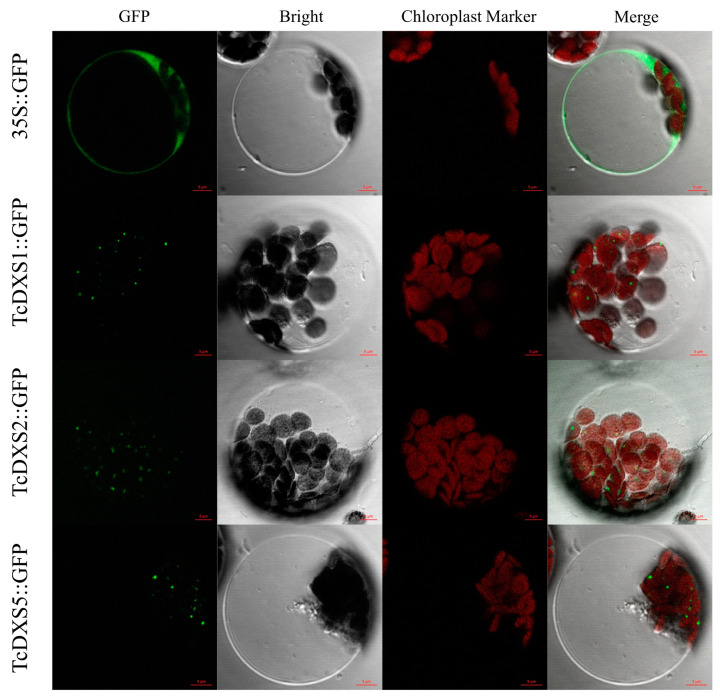
Subcellular localization of all TcDXS1, TcDXS2 and TcDXS5 proteins in the chloroplast envelope membrane of *A. thaliana*.

**Table 1 ijms-24-02339-t001:** Detailed characteristics of *TcDXS* genes of *T. ciliata*.

Gene Name	Gene ID	Strand	CDS (bp)	Protein			Predicted Subcellular Localization
				Length (aa)	PI	MW (kDa)	
*TcDXS1*	Tci11G005020.1	−	2145	714	7.93	76.83	Chloroplast
*TcDXS2*	Tci12G005860.1	+	2145	714	6.94	77.02	Chloroplast
*TcDXS3*	Tci21G002650.1	−	2166	721	7.03	77.74	Chloroplast
*TcDXS4*	Tci22G009550.1	+	2166	721	7.15	77.68	Chloroplast
*TcDXS5*	Tci25G002430.1	−	2187	728	7.11	78.77	Chloroplast
*TcDXS6*	Tci25G013250.1	+	2160	719	5.99	78.39	Chloroplast

## Data Availability

The data presented in this study are openly available in CNGB (https://www.cngb.org/, accessed on 11 February 2022) record number: CNP0001985.

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
