# Peer review of "Characterization of the 1-Deoxy-D-xylulose 5-Phosphate synthase Genes in Toona ciliata Suggests Their Role in Insect Defense"

_ijms, 2023, doi:10.3390/ijms24032339_

Round 1

Reviewer 1 Report

In general, this is a good article. I learned a lot from this article. Whether it is the author's propositional perspective, innovative theoretical views, or rigorous and realistic academic style and research ideas, people benefit a lot.

Reviewer 2 Report

The manuscript used bioinformatics analysis to identify TcDXS gene family in Toona ciliata and studed the molecular characterization of TcDXS gene family. Some issues should be addressed before its publication. 

1.         Figure 1 should be simplified to only show the differential sequence focused by the manuscript.  

 2.         The analysis is performed on the whole genome of T.ciliata in Line 13. The identified TcDXS genes are located in chr11, chr12, chr21, chr22, chr25. However, Table 1 in Line 191 shows that TcDXS genes are mainly in Chloroplast. This point is suggested to be discussed.

 3.         The research progress on function of gene families is suggested to be extensively introduced. For example, https://doi.org/10.3390/plants11020146, https://doi.org/10.3390/plants11060736.

4.         In Line 221, Figure 1 should be corrected to Figure 7.

 5.         There are some errors for the references. For example, lines 108, 111,120,134,148,179.

 6.         The English should be improved. For example, 'indicate' should be 'indicates' in Line 196., 'provenance' should be 'provenances' in Line 221.

Reviewer 3 Report

I am afraid that this manuscript is unsuitable for publication in its present form, since it looks and feels like a very rough first draft that needs a lot of work before it is ready for the review process (Were the authors perhaps under pressure of some deadline?). I confess that, after noticing some very obvious "red flags" (as described below), I have not read the rest of the paper very  carefully, and there may be additional major problems. Nevertheless, before the authors consider a possible resubmission, they must address the following critical issues:
1) All Figures and tables must be referred to from the text (and, while this is a "mere formal problem", they also should have unique numbers, to make orientation in the paper somewhat possible - there are currently two instances of Fig. 1 and no Fig. 7). Currently, only three of the 8 figures are mentioned in the text, and there is also no reference to Table 1. Moreover, there are multiple error messages indicating lost references to either figures or literature sources ("Error! Reference source not found").I would also strongly recommend placing the figures into the corresponding subchapters of Results instead of making a separate chapter consisting only of tables and figures.
2) The description of experimental work in Results, Materials and Methods, Figures and the Figure legends must be internally consistent, which is  not the case.
- Figure 8, which clearly shows images of protoplasts, is described as "subcellular localization ... in N. benthamiana leaves", text in Results does not provide any information about the experimental procedure (or even reference to this figure), and Materials and Methods speak of A. thaliana protoplasts.
- The scale of all the gene expression figures looks like some kind of normalization to value 1 for the first sample/control, but Methods state that ratio to multiple reference genes was used. How were these values calculated, after all?
3) The description of experimental and analytic procedures must provide sufficient information to allow reproduction of the results, which is currently not the case. To take the bioinformatic/phylogenetic part of the study as an example, the following essential information is missing:
- Which sequence was used as the query for BLASTP searches, which databeases were searched, and what was the E-value cutoff (or other inclusion criteria)?
-  In general, all references to large multitool packages such as TB Tools, MEGA or Jalview must be accompanied by information on specific tools, settings and algorithms used (e.g., which of the algorithms available within the JalView package has been used to construct the multiple sequence alignment?).
- How was the alignment processed prior to phylogenetic tree calculation, and how were gaps treated during tree construction?
- How was the phylogenetic tree of promoter sequences (Fig. 5A) constructed??
4) References have to be provided for authors´ previous observations described in Discussion (l. 231-245).
5) The author´s observations have to be discussed in the context of previous published work - especially regarding the phylogeny. Is the author´s phylogeny consistent with previous reports on the same gene family? If yes, this has to be clearly stated, if not, possible reasons have to be discussed. In any case, the authors should make clear what is the relationship between their gene/protein terminology and that used in previous publications. In particular, including in table S2 also previously used gene names, database accessions and standard locus identifiers would be extremely helpful.
6) From Fig. 8, it is not really clear where the GFP foci are located with respect to chloroplasts. Higher magnification photos have to be provided and empty vector, as well as non-transformed protoplasts, have to be shown. What do the authors mean by "dorsal side of the chloroplasts"?? This is not a commonly used term, and, while it might refer to the dorsal leaf side, IMHO it makes no sense in the protoplast context.
Last but not least, although I do not feel qualified to judge the author´s English, not being a native speaker myself, I had some difficulties understanding what are the authors trying to say at some places, and I believe that the paper would benefit from a language check.

Round 2

Reviewer 3 Report

Since the first version, this paper undervent major positive changes and reached a stage where it can undergo a detailed review. It actually turns out to be a well-performed, quite data rich study that is scientifically sound and may provide background for further applied experiments. Nevertheless, some issues regarding data presentation are still remaining, and there are also several places where the text would require some modifications for clarity, and I therefore suggest another round of revisions. I am attaching a commented version of the manuscript detailing the above mentioned issues, as well as additional comments and suggestions that mostly fall into the "minor or formal issues" category (but there are quite many of them).
1) I suggest including some reference to insect defense in the title - perhaps renaming the paper to "Characterization of the 1-deoxy-D-xylulose 5-phosphate synthase genes in Toona ciliata suggests their role in insect defense"
2) The abstract would benefit from some reorganization, shortening and deleting unimportant details, especially the numerical values of gene length or protein MW (see attached file).
3) The terminology of terpene categories (l. 46-49) is obviously incorrect - C5 compounds are not sesquiterpenes! Also consider whether this categorization is necessary and relevant in the context of this paper.
4) There is still an error code instead of reference to Table 1. I suspect that the error code may have been generated upon upload, so the authors should carefully check that the manuscript has been cleaned of all Word macros and field codes before submission.
5) Table 1 and all figure legends must be typeset in such a manner that they are not broken by page breaks.
6) Some background for the proposed renaming of genes (l. 108-109) must be provided, especially the previously used names (probably in Table 1, together with references where they were used).
7) Automatically computed values of CDS length, MW, pI and similar are of very limited, if any, biological relevance and, in my view, clearly do not deserve to be described in a redundant and repetitive manner. Please remove redundancies between the text (l. 110-117) and Table 1.
8) Please make sure that all lettering in all figures (including axis descriptions etc.!) is at least 8 pt at print size. Especially Figure 1 is too small to be legible - I suggest making it a Supplementary Figure, because otherwise it would have to be a multi-page one. Some parts of Figures 4 and 5 are also too small, I would suggest splitting these into several separate figures that can have lettering large enough to be legible. Parts 4A and 5A (phylogenetic tree) could be perhaps left out because they are redundant with Fig. 2.
9) Please provide either in the legend to Fig. 1 or in a supplementary file (mentioned in that legend) accessions of all the sequences shown in the alignment (aren´t they the sequences from the current Table S2?).
10) Supplementary tables should be numbered sequentially, which is currently not the case (i.e. the list of genes should be S1 and list of plant accessions S2, not vice versa).
11) Chapter 2.5 contains too little information to stand on its own. I suggest including its contents at the beginning of the current chapter 2.8 (and renaming the subsequent chapters accordingly).
12) The method used to measure gene expression should be called "reverse transcription - quantitative PCR" (RT-qPCR), not "quantitative real time  PCR" (qRT-PCR) - see https://doi.org/10.1042/BIO20200034
13)  Figure 8 appears to be blurry (too much compression in Word?). Please check resolution and settings of your editor, I recall it being much better in the previous version, i.e. showing more detail than now!
